# Pharmacogenomics at the Point of Care: A Community Pharmacy Project in British Columbia

**DOI:** 10.3390/jpm11010011

**Published:** 2020-12-24

**Authors:** Samantha Breaux, Francis Arthur Derek Desrosiers, Mauricio Neira, Sunita Sinha, Corey Nislow

**Affiliations:** 1Pharmaceutical Sciences, University of British Columbia, Vancouver, BC V6T 1Z3, Canada; sbreaux@mail.ubc.ca (S.B.); maunei001@gmail.com (M.N.); sunita.sinha@ubc.ca (S.S.); 2British Columbia Pharmacy Association, 430-1200 W. 73rd Avenue, Vancouver, BC V6P 6G5, Canada; derek@dessonconsulting.com; 3Sequencing and Bioinformatics Consortium, Office of the Vice-President, Research & Innovation (VPRI), University of British Columbia, Vancouver, BC V6T 1Z3, Canada

**Keywords:** community pharmacy, pharmacogenomic testing, pharmacogenetics, genetic privacy, pharmaco-economics

## Abstract

In this study 180 patients were consented and enrolled for pharmacogenomic testing based on current antidepressant/antipsychotic usage. Samples from patients were genotyped by PCR, MassArray, and targeted next generation sequencing. We also conducted a quantitative, frequency-based analysis of participants’ perceptions using simple surveys. Pharmacogenomic information, including medication changes and altered dosing recommendations were returned to the pharmacists and used to direct patient therapy. Overwhelmingly, patients perceived pharmacists/pharmacies as an appropriate healthcare provider to deliver pharmacogenomic services. In total, 81 medication changes in 33 unique patients, representing 22% of all genotyped participants were recorded. We performed a simple drug cost analysis and found that medication adjustments and dosing changes across the entire cohort added $24.15CAD per patient per year for those that required an adjustment. Comparing different platforms, we uncovered a small number, 1.7%, of genotype discrepancies. We conclude that: (1). Pharmacists are competent providers of pharmacogenomic services. (2). The potential reduction in adverse drug responses and optimization of drug selection and dosing comes at a minimal cost to the health care system. (3). Changes in drug therapy, based on PGx tests, result in inconsequential changes in annual drug therapy cost with small cost increases just as likely as costs savings. (4). Pharmacogenomic services offered by pharmacists are ready for wide commercial implementation.

## 1. Introduction

Completion of the Human Genome Project in 2003 brought expectations that the information would revolutionize the practice of medicine and introduce new scientific, business, and medical models [1,2]. While many of those hopes are just beginning to be realized, the resulting discipline of pharmacogenomics (PGx) has matured considerably in the past decade. PGx uses genetic information to classify patients who may benefit from personalized medication or who may respond negatively to a particular treatment. PGx can help ensure that patients receive the most appropriate medication and dose, can reduce the number of adverse drug reactions (ADRs) and aid in medication adherence. The most appropriate provider of PGx testing, however, remains a subject of debate. In British Columbia (BC) Canada, pharmacists are the recognized drug experts [3]. Furthermore, over the past two decades their scope of practice has expanded to provide more aspects of comprehensive patient care [4]. These additions are a powerful way to address the fact that every year in BC over 200,000 people are admitted to hospitals due to adverse drug reactions of which 10,000–20,000 die; these patients’ treatments cost an estimated $49 million per year [5]. These numbers are likely to be higher because 95% of ADRs go unreported [6]. In 2011, the American Pharmacists Association acknowledged the importance and practicality of integrating genomics with medication therapy management programs to optimize patient drug therapy [7]. Such emphasis on a more patient-centered, individualized, and preventative approach to wellness is an antidote to the frustration of the one-size-fits-all paradigm of evidence-based medicine [8]. Implementation of PGx testing based on these benefits has, however, proven to be challenging. Causes include low acceptance of pharmacist recommendations by the physician and prescriber, mixed patient receptivity, low rates of reimbursement to pharmacists, inadequate human resources, and the physical layout of the pharmacy [9]. Our supposition for potentially unproductive interactions between pharmacists and physicians was due to the (self-reported) high levels of unfamiliarity with regards to genomics and by extension being uncomfortable with making drug therapy changes based on a participant’s drug metabolism genotype [10]. An additional barrier is the cost of PGx testing which ranges from $200–$500, often left to the consumer because insurers have been hesitant to cover genetic testing for non-diagnostic purposes [11]. Fears include concerns over data security and actual clinical impact [12]. These barriers are surmountable and have been addressed in other contexts [13].

Building on earlier work in which we concluded that the community pharmacist is the appropriate healthcare expert for PGx deployment [14,15,16]; in this study we tested the hypothesis that medication changes as a result of PGx testing have a minimal impact on the overall cost of a patient’s drug therapy. In today’s market, there is a diversity of PGx-testing platform technologies [17]. DNA arrays and polymerase chain reaction (PCR)-based tests are commonly used methods for commercial genotype screening. An advantage of these two assays is that they are largely blind to detecting so-called incidental findings. Specifically, both arrays and PCR are used to confirm either the presence, absence, or duplication of specific known single nucleotide polymorphisms (SNPs) and as a result only information about those alleles under study can be gleaned from this process. Furthermore, the accuracy of these approaches has been validated [18,19,20,21] and they are simple and cost-effective, making them easy to implement in routine practice.

The objectives of this study were to; (1) test the feasibility and appropriateness of community pharmacists as a conduit for pharmacogenomics information, (2) to gauge the receptivity of patients in this setting and (3) assess the cost-effectiveness of this approach. Despite the limited size of the study, we satisfied these objectives and discuss how the lessons learned here can be applied.

## 2. Materials and Methods

See Appendix A.

### 2.1. Pharmacy Selection

Community pharmacies were selected to reflect a diversity in geography and practice environments in BC. Pharmacies were required to have expressed interest in participating, a corporate membership with the BC Pharmacy Association, a sufficiently private counselling area and adequate staffing to ensure that the pharmacist could have uninterrupted time with participants during the education and consent process. Additional pharmacies were added as needed. Accounting for individual turnover, we ended up with 21 pharmacists recruiting patients at 17 participating community pharmacies in 13 locales across the province as shown in Map 1.

### 2.2. Pharmacist Training

In addition to the Tri-Council Policy Statement Ethical Conduct for Research Involving Humans Course on Research Ethics, the pharmacists had to complete a study training program done remotely via webinar and phone; (i) to ensure pharmacists followed all the requirements of the law and Research Ethics Board of the University of British Columbia (UBC), especially with respect to patient privacy and (ii) to ensure that the patient experience was consistent regardless of the pharmacy type or location.

A study team member and the pharmacist discussed the project principles of informed consent, privacy requirements, patient education, obtaining consent, collecting patient information, and reviewed a consent checklist designed to guide the education and consent process. At the conclusion of this session, the pharmacist was asked a series of questions based on the training they received.

### 2.3. Operations Logistics & Report Interpretation

The details of sample collection, handling, return, and documentation were discussed with a study team leader. Pharmacists were required to complete the myDNA online pharmacist training program for PGx as well, providing an overview of pharmacogenomics as well as interpretation of the myDNA reports. The learning objectives for this training were (1) understand the basis of cytochrome (CYP) P450 genes/enzymes associated with CPIC guidelines, (2) understand how variants affect an individual’s ability to metabolize medications, and (3) how to apply this knowledge in clinical practice to improve their patients’ outcome.

### 2.4. Quality Control (QC)

Before the pharmacists enrolled patients in our study, a phone call to role play the registration and consent process with a study team member was conducted. The study team member completed the consent checklist (Appendix A) during the process and at the end of the session reviewed the terminology, phrasing, and content with the pharmacist.

### 2.5. Participant Selection and Consent

To be enrolled in the study a potential participant must have been over 19, speak English, and needed to be taking a valid criteria drug at time of enrollment. Pharmacists were prohibited to search patient records to identify eligible participants. In a private area of the pharmacy, the pharmacist explained the project and summarized the Participant Information & Consent Form (Appendix A). A checklist was completed for each potential participant. The potential participant was then shown a video specifically developed for this project. The video, (Appendix A), introduced the key concepts of PGx and the goals of the research project. The pharmacist watched the video with each patient to ensure that concepts were clear and to answer questions as necessary. The potential participant was then given the Patient Information & Consent Form to review, and was required to wait at least 24 h before committing to the study. This allowed patients time to reflect, to discuss the project with other family members or caregivers, and to obtain additional information to make an informed decision about their participation.

After a potential participant agreed to the study, the enrollment process started with the pharmacist answering questions generated in the contemplative (take-home) phase. Next, patients signed the consent form and were given a copy for their records. Following their consent, the patients provided a saliva sample (see Appendix A) and their pharmacist collected the required enrollment information. To avoid external incentives (or the appearance thereof) we specified that each pharmacist be limited to recruiting a maximum of 10 patients.

### 2.6. Data & Sample Collection

Mandatory information collected included date of birth, gender, current medications, history of ADRs as well as allergies and medical conditions. Disease and indication data were not collected from participants. Even though gender, age and drug therapy information were collected, the numbers in the study were too small to sufficiently address stratification by any of these data. The Genotek Oragene saliva collection kit was used according to manufacturer’s protocol to collect patient sputum [22]. This process took 2–5 min in most cases, although there were participants who took longer and a small number who were unable to provide usable saliva samples. The reasons for this varied, but the common theme was that these participants complained of ‘dry mouth’.

### 2.7. Experience Survey

A pharmacist and patient experience survey were mailed out with the recruitment kit (Appendix A). The enrolling pharmacist ensured completion and return of the surveys at the end of the study. They were asked to indicate their level of agreement using a 4-point Likert scale, which was chosen over a 5-point scale [23], removing a “Neutral” option to require respondents to either agree or disagree with the statements.

### 2.8. Transport of Samples & Participant Information

After de-identification, the original copy of the patient enrollment documentation and the patient’s saliva sample were sent via secure courier to UBC. A copy of the demographic information was kept and secured at the pharmacies. Saliva samples were received and catalogued and stored at our sequencing facility (https://sequencing.ubc.ca/). Participant information was used to update a key file linking identifying information to the participant code. All non-identifying information was transcribed and linked only to the participant code. Sample IDs were then subsequently linked to unique, randomized sample barcodes for downstream analysis and report tracking.

### 2.9. Sample Processing

DNA was extracted from 250 µL of saliva sample. Any remaining saliva was stored at room temperature for up to a week prior to long term storage at −20 °C. The “prepIT.L2P” reagents were used according to the manufacturer’s instructions (DNA Genotek). DNA was eluted in 50 µL molecular-grade water and DNA quality was assessed by gel electrophoresis and quantified by Nanodrop (Thermofisher Scientific, Waltham, MA, USA) and fluorometry using the Qubit dsDNA HS Assay Kit. The gel analysis provided a go/no-go step for the samples, in other words, if samples were extensively degraded at this QC step, we attempted a second extraction. DNA was stored at −20 °C until genotyping or TRS library preparation.

### 2.10. TargetRich Sequencing (TRS)

DNA was extracted as described above and processed according to the manufacturer (https://www.kailosgenetics.com/). Briefly, to prepare the sequencing library, guide oligos which contain the sequences to be amplified are annealed, followed by a restriction enzyme digestion, after which Illumina adapter sequences are annealed along with the unique identifier (barcode) for the library sample. The samples are then enzymatically cleaned via magnetic beads before being amplified and cleaned a final time. Samples were QC’d by agarose gel electrophoresis and quantified with Qubit. Pooled amplicons were sequenced on an Illumina Miseq platform, generating paired-end 78 bp reads [24]. Long range PCR was used to determine duplication as described by the manufacturer [25].

### 2.11. Genotyping

We worked with myDNA—https://www.mydna.life/en-ca/to perform SNP analysis using the iPLEX MassArray System, a non-fluorescent platform utilizing MALDI-TOF (matrix-assisted laser desorption/ionization—time of flight) mass spectrometry, coupled with end point PCR to measure PCR-derived amplicons in multiplexed reactions. Briefly, polymorphic sites were detected by primer extension where the targeted region is amplified; remaining dNTPs are neutralized and then a terminating extension reaction using a promoter that binds immediately upstream of the polymorphic site as a ‘mass modified’ nucleotide lacking the 3′-hydroxyl extends the product by a single base [19,20,21,26]. The number of CYP2D6 gene copies was detected by qPCR using a 7900HT PCR system [27].

### 2.12. Data Reporting

Patient reports were generated using myDNA’s PGx software (https://www.mydna.life/en-ca/). These reports were uploaded to a secure website accessible to the primary project team by the PI (CN), the User Partner Lead (FADD), and the project’s Research Assistant (SB). Data was encrypted and only de-identified to the appropriate pharmacist after review by the project team. Genomic reports and patient IDs were sent separately in encrypted Excel spreadsheets. GitHub (https://github.com/) was used to store all analysis routines and to ensure version control.

In addition to genotyping 150 samples, 46 were subjected to Kailos TRS or “target rich sequencing protocol”. The NGS data and the final TRS reports were not returned to the pharmacists and restricted to internal comparisons.

### 2.13. Patient Consults at the Pharmacy

Every patient enrolled in the study who was able to be genotyped received a copy of their myDNA report. Neither the patient nor the pharmacist was returned a copy of the TRS report, which was used for our own reference to further validate the myDNA results as well as do a basic comparison of the functionality of sequencing over an array-based analysis. The reports were released directly to CN and FADD at which point we would review them before informing the pharmacist. Reports were reviewed with each participant in a face-to-face appointment with the pharmacist following a standardized script. The pharmacist delivered results, discussed possible therapy change recommendations, and asked if the participant wanted the report shared with the patient’s physician. Participants had the option of sharing the report directly themselves or having the pharmacist send a copy. Pharmacists were responsible for recording medication changes. All medication changes were made the patient’s physician or general practitioner and all participants were asked to complete a qualitative survey.

### 2.14. Data Collection & Analysis

To process the myDNA reports for our meta-analysis, each participant’s medical considerations and genotypes were extracted from PDFs using tabula [28]. Files were then manually edited to include a patient ID and any potential drug-drug interaction information. Genotype information from the TRS reports were manually entered into a .csv file and further tidied, such as conversion from wide to long data, using R (version 3.6.1), a programming language for data analysis [29]. To compare genotype calls between TRS and myDNA, only shared alleles were analyzed. A file containing every unique myDNA call was matched with the corresponding TRS genotype. Population frequencies for the genotypes CYP2D6, CYP2C19, CYP2C9, and VKORC1 were taken from an analysis of an Australian population [27]. The frequency of CYP2D6 *36 was taken from an American population [30]. The population frequencies of the SLCO1B1, CYP1A2, CYP3A4, CYP3A5, and OPRM1 genotypes were calculated from the global SNP frequency. Global Frequency of the SNPs were gathered from the Genome Aggregation Database (gnomAD) (https://gnomad.broadinstitute.org/) [31]. Hardy-Weinberg equilibrium [32] was used to calculate the genotype frequencies in an ideal population.

All genotype data manipulation and analyses were completed in R version 3.6.1 (Appendix A). Analysis depended on R packages: Tidyverse, data.table, reshape2, compare, plyr, and rowr [33,34,35,36,37,38]. Cost-benefit analysis and tabulation of survey results was completed in Excel. Drug prices were retrieved from the McKesson Canada wholesale drug price list in effect at that time.

### 2.15. Research Ethics Board Approval & Legal Compliance

In developing our Research Ethics Board (REB) procedure, we considered the following Canadian and British Columbian legislation:The Personal Information Protection Act, The Freedom of Information and Protection of Privacy Act, The Health Professions Act and its Bylaws, The Health Care (Consent) and Care Facility (Admission) Act, and The Pharmacy Operations and Drug Scheduling Act. These laws lay out the obligations of the pharmacist, the pharmacy and the University of British Columbia with respect to personal and health information.The Health Professions Act and its Bylaws and The Personal Information Protection Act. These laws governed the pharmacist with respect to the collection, use, disclosure and security of personal and health information.The Freedom of Information and Protection of Privacy Act and the policies of UBC and its Research Ethics Board.

### 2.16. Timeline

Starting in mid-2017, our project ran until January of 2018. Most aspects of the project were completed in tandem as opposed to sequentially. In the first six months we prepared the pharmacist training material and updated patient recruitment kits. During this time, we initiated patient marketing and recruitment. While the bulk of these activities was completed in the first 6 months, recruitment persisted until completion of sequencing. Enrollment began after pharmacist training was complete and patients had to be taking at least 1 of the mental health drugs listed in Table 1. We used a batch approach to sample processing, beginning 6 months after project initiation and persisted for an additional 9 months. Data analysis began after first results were returned in quarter 3. The last activity we accomplished were the pharmacist consultations where we returned reports and completed the final aspect of our data analysis. These activities persisted for 9 months.

## 3. Results

In this study we built on our and others work to further test feasibility of community pharmacogenomic testing, in addition to assessing pharmacist and community comfort with pharmacogenomic services and to conduct a simple drug cost analysis [14,16,39]. To accomplish this, 21 pharmacists at 17 pharmacies, Figure 1, were enlisted to recruit 150–200 patients for genotyping (or genotyping and TRS) when they filled or renewed a prescription for an antidepressant/antipsychotic, Table 1. MyDNA genotyping analysis (https://www.mydna.life/en-ca/) was used to assess patient responses to a wide variety of medications with a focus on mental health medications.

Antidepressants and antipsychotics are metabolized by diverse enzymes. The cytochrome p450 isoforms CYP2C19 and CYP2D6 are responsible for metabolism of more than two-thirds of the currently available psychiatric drugs; these genes are also highly polymorphic with a variety of stable alleles and mutations, including whole and partial duplications and deletions [40,41,42]. As a consequence, the range of enzyme activity and downstream phenotypes is large [40]. Indeed, the amount of clinically relevant mutations in these genes appears to be above 50% for most populations [40]. Additionally, the rate of initial response to antidepressant treatment was only 49.6% [41]. The additional costs incurred for management of these non-responders is ~$10,000USD/yr./patient [43]. This combination of factors; (1) a large pool of diverse alleles, (2) high degrees variation in drug metabolism and the high costs of productive patient prescribing highlight the importance and usefulness of personalized treatment for these medications [42]. Actionable results (based on up-to-date guidelines from the Clinical Pharmacogenetics Implementation Consortium (CPIC) [44] were returned to the pharmacist for review (see a sample report- Appendix A) with the patient and the prescriber (if appropriate). Net costs were calculated for all therapy changes made to a patient’s current medications. Patient and pharmacist experience surveys were used to judge the participant’s thoughts on the services and experience. We also assessed if the medication was discontinued/changed/dosage altered, the overall financial impact of the changes on drug therapy costs, and the reliability and quality of genotyping results.

Sample collection and genotyping was accomplished in two main batches. Batch one comprised 130 samples, 116 of which passed QC. In batch two 48 samples were collected, 42 of which passed QC. We also received 19 samples as a retest, in total generating 150 myDNA and 37 TRS genetic reports, with 47 samples that did not pass QC. For example, some patient’s sputum simply did not provide adequate DNA as re-extraction only continued to produce insufficient or degraded samples. This may be due to an inability to produce the appropriate amount of sputum, natural variations in cheek shedding, or effects of medications.

### 3.1. Comparison of Genotypes

We found 9 total differences in genotype calls between those that underwent both TRS and myDNA genotyping (for a total of 592 SNPS), Table 2. Between the two datasets there were 296 comparable genotypes giving a discordance of 1.7%. One gene could not be called by TRS. This may have been due to the region being degraded or problems with amplification for the patient. TRS also called two additional alleles that myDNA does not, CYP2D6 *35A and CYP3A4 *8. *35A is a subset of the *2 allele. *2 contains SNPs 2851: c > t and 4181: g > c, while *35A contains the additional SNP 31 g > a. *35A has the same normal metabolizer phenotype [45]. As such, the two calls containing *35A can be considered the same as that by myDNA. The CYP3A4 *8 allele has been associated with decreased function of the CYP3A4 protein, although PharmGKB lists this as a level 3 (i.e., low evidence) claim [46]. Regardless, this genotype is absent in the myDNA report resulting in a normal metaboliser call. The remaining differences were minor, suggesting a small number of SNP-specific variables for each platform.

Next, we compared the frequency of a subset of genotypes that were part of both the TRS and myDNA reports. Genotypes were compared both to each other and to the population average. Population averages, comprising of Australian, American, and global ethic data [27,30,31] closely matched those from within the study at both sites, Table 3. The averages between myDNA and TRS were similar, showing little variance between the two data types.

### 3.2. Community Acceptance

To gauge the scope and scale of community acceptance a simple two-pronged quantitative, frequency-based analysis of patient and pharmacist attitudes and thoughts was conducted via simple surveys. Each participating patient was asked to complete a short seven question survey in which they ranked their response to statements about the project. Similarly, each participating pharmacist was asked to complete a survey in which they ranked their response to statements about the training and support they received throughout the project. We received 20/21 pharmacists’ experience surveys and 111 patient experience surveys with a response rate of 62%. Some patients were not able to be reached at the end of the study and one pharmacist dropped from the study. The patients strongly agreed with the seven statements and also agreed that pharmacists are the appropriate providers of pharmacogenomic services as well as pharmacies being an ideal location to collect samples, Figure 2.

Pharmacists’ opinions were generally very positive as well, Figure 3. The biggest pharmacist’s concern was communication with our research team. This is a fair criticism and likely reflects two constraints of the experimental design; (i) because samples were batched, an overly long time (up to six months) between sample collection and report returns was experienced for the samples collected earliest in the project, and (ii) the project team strove to maintain an arm’s length distance for any prescribing decisions.

### 3.3. Drug Cost Analysis

The myDNA reports returned to the pharmacists were used to produce the data in the drug cost analysis. Reports offered three prescribing considerations: ‘usual—normal label use of compound’; ‘minor- consider test results, as results may be significant’; and ‘major—significant results, medication should be reviewed’. The restriction to mental health drugs was only for the eligibility to participate. Once a participant was enrolled, we reviewed all their drugs and many of the drug therapy changes that were made were for drugs other than mental health drugs. All drug changes, regardless of therapeutic category, were included in the simple drug costs analysis. In a small number of cases (16), reports could not be returned as some pharmacists had lost contact with study participants. Additionally, some doctors either felt uncomfortable changing prescribing considerations based on the report results or did not think it was necessary for some patients. For medications that patients were currently taking, 92 were found to have at least one minor prescribing consideration, 39 had at least one major consideration, and an additional 139 participants were taking a medication with usual prescribing considerations, Figure 4. In comparison to a PGx study examining 3 genes using a WES data set, 20% of study participants had immediately actionable results, comparable to the 26% that we found with a major prescribing concern [47].

Taken together, the aggregate medication changes translated into therapy interventions in 33 patients, representing 22% of all genotyped patients in the project. In addition, the report interpretation with the pharmacist and participant often prompted closer review of patient medications by physicians. There was a total of 81 changes. The changes included dose increases in 11 patients, dose decreases in 5 patients, new drugs added to the therapy of 20 patients, and 22 patients having drugs discontinued. There were instances of multiple changes for an individual patient, Figure 5. Based on this data, we calculate that a year’s worth of modified medication therapy for all participants collectively was $797CAD. This represents a per patient cost of $24.15CAD (annual drug cost based on patient specific dosages and net of all changes including discontinued drugs, new drugs added and/or dosage changes) considering only those patients who had a medication change (not including the initial non-recurring testing cost of $199 which was covered by the project budget and should be amortized over the life of each patient). Note that costs in this simple drug cost analysis are all based on annual ongoing treatment costs and are not limited to the actual prescription over the study period. That is to say, the per patient cost of $24.15 represents the average total annual cost increase for each patient’s therapy after implementing the changes. It is not restricted to only the cost of each patient’s therapy for the study period. Study participants were not followed at all beyond the consultation with the pharmacist to review results and implement any suggested drug therapy changes. This was a time and budgetary limitation of the study.

## 4. Discussion

PGx testing is a cost-effective service valued by the participants. More work should be pursued to further educate physicians and drug/insurance providers on the benefits and potential improvement to patient treatment outcomes and well-being to enhance acceptance and implementation in BC. Additionally, prescribers need further education on PGx concepts. While with relatively simple courses our pharmacist felt confident in their understanding of the science and rational behind PGx testing—Doctors may feel uneducated to make prescribing changes based off of PGx information [10,48]. This information may contain contradictory or poorly validated results that could lead to denial of treatment [49]. However, integrating pharmacists as the drug experts to guide PGx prescribing, creating standardized reporting guidelines, and educating clinicians promises to improve the reliability of PGx dosing.

While the results that have come from this project might allow us to extrapolate to a large number of very specific conclusions related to PGx testing in the community by pharmacists, we have limited our conclusions to the following six statements:The public perceives pharmacists/pharmacies as a very appropriate healthcare professional/venue to deliver pharmacogenomic services.Frequencies of alleles, interactions, and clinically actionable results are consistent with other studies published in the scientific literature.Changes in drug therapy based on PGx test results represent an inconsequential change in annual drug therapy cost. While drug therapy changes may result in a small cost increase, it is just as likely that costs may decrease.Any cost increase due to drug therapy changes is likely to be small and is justified on the basis that the patient will be taking the most appropriate drug and dose for them as an individual based on their phenotype.Pharmacogenomic testing is appropriate and affordable for certain patient populations.Pharmacogenomic services offered by pharmacists are ready for primetime wide commercial implementation.

### 4.1. Selection of Antidepressants/Antipsychotics as Inclusion Criteria

In consultation with one of our funders, Green Shield Canada, we decided to focus on mental health drugs. Two out of three people will need to try multiple/different antidepressants until they find one that works for them [50]. While this may not match the amount of medication changes, we found (22%) we don’t know how long the patient has been taking their psychiatric medication and if they are satisfied with the results of them. Antidepressant/antipsychotic usage was a criterion for the study patients may have had their own personal reasons for choosing to enroll. We also don’t know how many different antidepressants they’ve been on previously. Additionally, they may not be taking them for their major indication but rather an off-label effect. The most common reason for needing to switch was due to side effects, which can leave a person physically debilitated and even worsen their mood disorder [50]. Additionally, antidepressant use is linked with age, with the elderly (those over 60) being 40.2% more likely to use an antidepressant than the rest of the population [51]. The elderly also take multiple classes of drugs with 51.6% of seniors in Canada taking 1–4 drugs of different classes chronically and an additional 35.3% taking 5 or more [52]. Some of these drugs are used to mitigate ADR symptoms from their other medications. Identifying problematic medications can reduce the drug cost if other medications can be discontinued if they are no longer needed to manage ADRs.

### 4.2. Pharmacist- & Pharmacy-Specific Considerations

We erred on the side of caution in making sure that the pharmacists had a high level of familiarity with PGx (equivalent to a 1st year graduate course), including its potential and its limitations. The quality, quantity, and level of detail of information provided in the individual patient reports generated in this project allowed pharmacists to easily interpret results and make drug therapy recommendations with little to no additional training. In BC, pharmacies are operated as private businesses with the ability to bill the public healthcare system for services. Using pharmacies as study sites required compliance with the privacy regulations specific to private businesses. In some instances, this was a higher threshold than that required by a public university research project. As the focus of this study was to develop and test a protocol that could be commercialized, we focused on ensuring compliance with the highest standards of privacy and informed consent. The underlying premise was that compliance, if introduced and explained at the outset with a clear rationale and requirements, would mitigate the potential for non-compliance. This was coupled with the idea that standardizing the process from the outset, would allow identification of any barriers present in each individual pharmacy practice setting. Participating pharmacists reported that the detailed training resulted in no difficulty in complying with the SOPs developed in Phase 1.

### 4.3. Potential Impact of S-201 and Other Pending Legislation

Patients were less concerned with privacy and confidentiality issues than we anticipated. Patients generally believed that pharmacists have access to their confidential health information, including their full medical record that exists with their physician. While this is not the case, pharmacists in the project were careful to ensure that patients understood the implications of sharing personal confidential medical information about themselves. Patients showed considerable trust in their pharmacists in handling this information and were pleased with the level of detail included in the project consent form. When this study was launched, there were no legal protections of a patient’s genetic information data. This changed in 2016 with the passage of bill S-201, the Genetic Non-Discrimination Act which provides robust, albeit untested, protections against discrimination based in genetic information [53]. In practice, we did not encounter resistance to participation but additional work will be required to assess the impact of these protections on patient behavior with regard to testing.

### 4.4. Drug Cost Consequences

Although the additional yearly per-patient cost is ~$25CAD, PGx testing represents a saving to the community as we maximize the therapeutic efficiency of treatments. In fact, other studies have shown cost saving benefits of PGx testing [54,55]. While opportunities in PGx are clear- reduction in ADRs, elimination of medication trial and error, and more accurate dosing of prescribed medications, data to support the economic argument of drug cost savings are limited. However, it is not a stretch to hypothesize and make an effective argument that an additional value of PGx testing is the avoidance of weeks to months of costly trial and error when prescribing multiple drugs, especially in the mental health realm. Thus, using PGx testing to get a patient on the right drug at the right dose has the potential to generate long-term savings relative to that patient’s overall healthcare costs. Furthermore, it could be argued that the wrong drug and/or wrong dose for a patient may contribute to poor adherence, further contributing to unnecessary costs. Using PGx could and probably does contribute to improved adherence, which in turn improves cost effectiveness of therapy. Longer-term economic implications related to reduced physician and urgent care (e.g., emergency room) visits, reduced absenteeism, and improved productivity require further study and analysis.

## 5. Limitations of the Study

While our study demonstrated the feasibility of pharmacist-led, community pharmacy-based pharmacogenomic testing there are several limitations. Although we attempted to, whenever possible, make the methodology suitable across Canada, there are province-specific considerations that will likely need to be considered. We also note that the size of this study is not sufficient to draw general conclusions regarding particular gene-drug interactions. It is also worth noting that although eligibility criteria included a limitation to being on at least one mental health drug, the gene-drug interactions reviewed in the reports included all relevant gene-drug interactions for each patient and not just their mental health drugs. Also, our decision to batch samples for processing slowed the return of results. Finally, in an effort to avoid potential privacy concerns, we did not collect detailed demographic data, nor did we follow the patients once the study was completed.

## Figures and Tables

**Figure 1 jpm-11-00011-f001:**
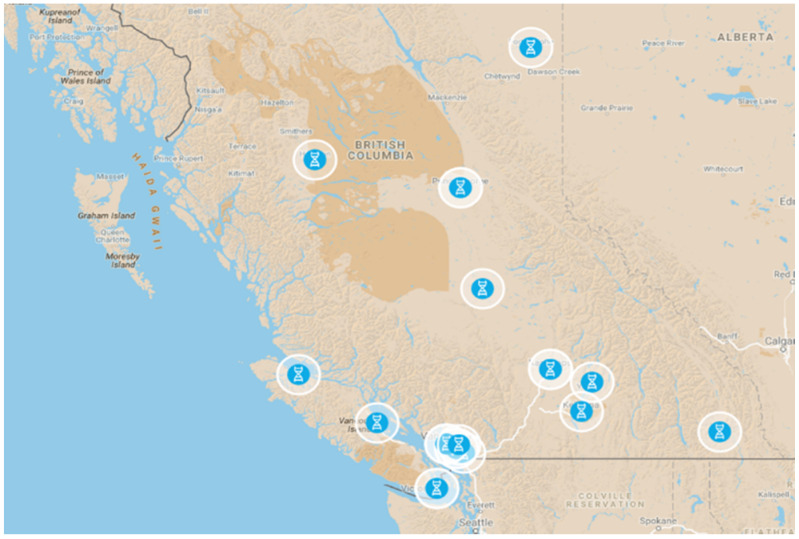
Map of participating pharmacies by their locations. For table of locations and pharmacies see Appendix A.

**Figure 2 jpm-11-00011-f002:**
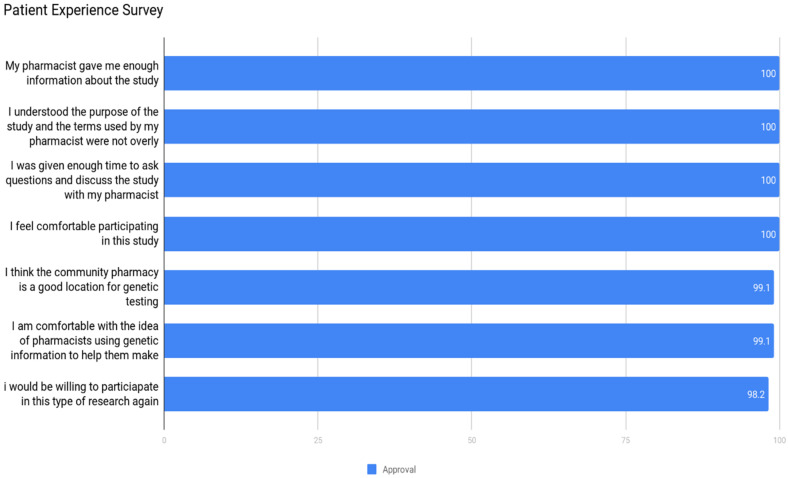
Patient experience survey. All results, 98.2–100% strongly agree/agree, *N* = 111.

**Figure 3 jpm-11-00011-f003:**
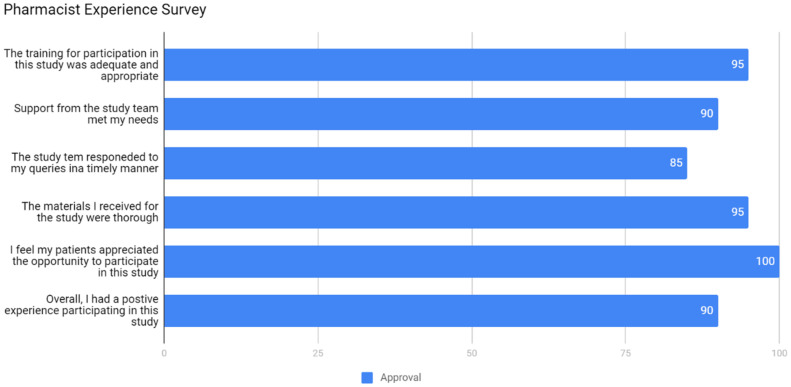
Pharmacist experience survey, 85–100% strongly agree or agree, *N* = 20.

**Figure 4 jpm-11-00011-f004:**
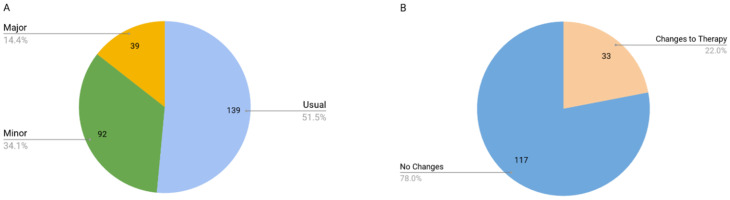
(**A**) Visualization of major, minor, and usual drug considerations discovered, *N* = 150; (**B**) Visualization of medication changes in response to the study, *N* = 150.

**Figure 5 jpm-11-00011-f005:**
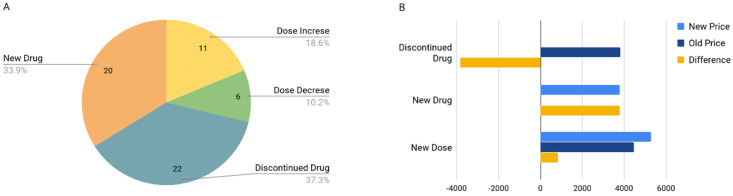
(**A**) Breakdown of therapy changes made by type of change, *N* = 59; (**B**) Cost-benefit of drug changes—shows drug cost changes by type of therapy change. Bars represent total cost in CAD.

**Table 1 jpm-11-00011-t001:** Study compounds. Patients had to be currently taking at least one of the medications in the table to be included in the study. Included are usage frequency of each drug. Some patients were taking multiple compounds.

Antidepressants	Usage	Antidepressants	Usage	Antipsychotics	Usage
Agomelatine	0	Mianserin	0	Aripiprazole	9
Amitriptyline	12	Mirtazapine	12	Clozapine	0
Citalopram	26	Moclobemide	2	Haloperidol	0
Clomipramine	0	Nortriptyline	6	Olanzapine	5
Dothiepin	0	Paroxetine	2	Quetiapine	24
Duloxetine	10	Sertraline	17	Risperidone	4
Escitalopram	27	Trimipramine	0	Zuclopenthixol	0
Fluoxetine	12	Vanlafaxine	23		
Fluvoxamine	1	Vortioxetine	2		
Imipramine	1				Total: 195

**Table 2 jpm-11-00011-t002:** Differences found between genes shared in the TRS and myDNA datasets.

GENE	TRS Genotype	myDNA Genotype	Comparison
CYP2C19	*XX/*XX	*1/*17	Kailos no call
CYP2C19	CYP2C19 *1/*2	*2/*2	different
CYP2C9	*1/*3	*3/*3	different
CYP2D6	*2/*2	*2/*5	different
CYP2D6	*35A/*5	*2/*5	Kailos only allele *35A
CYP2D6	*35A/*4	*2/*4	Kailos only allele *35A
CYP3A4	*1/*8	*1/*1	Kailos only allele *8
SLCO1B1	T/C Het	T/T Wild	different
SLCO1B1	T/C Het	T/T Wild	different

*: allele.

**Table 3 jpm-11-00011-t003:** Sample of a table comparing the frequency of myDNA calls and Kailos calls to population averages of those genotypes. Full table contains 62 genetic variations. See Appendix B.

GENE	myDNA Genotype	TRS Genotype	Phenotype	myDNA Genotype Frequency % *n* = 150	TRS Genotype Frequency % *n* = 37	Population Level Frequency %
CYP2C19	*1/*1	*1/*1	Normal metabolizer	35.33	27	39.7
CYP2C19	*1/*17	*1/*17	Rapid metabolizer	33.33	37.8	25.80%
CYP2C19	*1/*2	*1/*2	Intermediate metabolizer	14	18.9	20.70%
CYP2C19	*17/*17	*17/*17	Ultrarapid metabolizer	2.67	2.7	0
CYP2C19	*2/*17	*2/*17	High intermediate metabolizer	8	8.1	6.20%
CYP2C19	*2/*2	*2/*2	Poor metabolizer	6.67	2.7	2.90%
CYP2C19	NA	*XX/*XX	NA	NA	2.7	NA
CYP2C9	*1/*1	*1/*1	Normal metabolizer	69.33	62.2	64.84%
CYP2C9	*1/*2	*1/*2	High intermediate metabolizer	15.33	21.6	20.38%
CYP2C9	*1/*3	*1/*3	Intermediate metabolizer	10	10.8	10.60%

*: allele.

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
