# Peer review of "Pharmacogenomics at the Point of Care: A Community Pharmacy Project in British Columbia"

_jpm, 2020, doi:10.3390/jpm11010011_

Round 1
Reviewer 1 Report
Intro - Overall the introduction seems very long. Usually introductions, are 1-2 paragraphs and set up the reason for the particular study for which your study is being conducted. For example... starting at line 78, seems like this content is discussion or results. There shouldn't be figures or tables in an introductions.
After line 78 - add a sentence describing the objectives of your study. i have a sense of the objectives / aims based on results but would be best to have clear statements of the objective / purpose of the project all in one spot before all the methods are laid out.
Content below line 78 - these paragraphs belong in the methods. Lines 78-84 goes in methods with sample / test information. Some of 85 -101 goes in pharmacy selection section, and some goes with patient and medication selection, cost determination.
line 63 - "Building on earlier work", should this earlier work be cited?
Method section is very long (7 pages?). Is all of this absolutely needed to understand how the study was conducted? Much of this section can be reduced. one to two pages.
1) You can say the project was approved by the research and ethics board according to Canadian and British Columbian legislation.
2) Informed consent was obtained. 3) inclusion criteria - age 19, English speaking, taking at least one medication in table 1.
3) How were patients identified?
Results: should stick to presentation of findings, state results that is it. all explanations go in discussion (line 326). Some sentences are methods (line 316) and some belong in the discussion (eg 320-322)
start with how many completed the study. then genotype findings, medication recommendations changes, then cost savings. Then put survey results.
Fig4b - please clarify the scale.
Discussion I would start with your six conclusions and then discuss the results that enabled you to come to those conclusions. Can you have paragraph on the challenges / limitations of the study?
Reviewer 2 Report
General:
This is a well-written study assessing cost-benefit prognosis of pharmacogenomics testing on metabolising enzymes and 20 antidepressant/antipsychotic drugs for 180 patients in Canada. The authors provide a thorough background and explanation on their study methodology and procedures, and hence provides a useful framework, which could be easily adapted and implemented in other places. Their conclusions such as on the general benefits of PGx testing, inconsequential changes in annual drug therapy costs based on PGx tests as well as on pharmacists as a useful provider of PGx services are important, but not surprising given previous studies in this space. Some of the results could benefit from a more substantial analysis and quantification. My main concern is general confounding by disease, indication, treatment, sex or age, which is not sufficiently addressed. Furthermore, although clearly not a focus here, it would be interesting to provide more evidence and insights on a per drug basis and a follow-up outcome analysis on the patients, which received PGx test-based alternative treatment regimen.
The authors confirm their initial hypothesis that “medication changes as a result of PGx testing have a minimal impact on the overall cost of a patient’s drug therapy.“. Although their conclusions are in line with previous studies, the underlying calculations are not clearly presented. Specifically, Breaux et al. should elaborate and clarify on the following points:
- List a more detailed calculation on a per drug basis, a list of costs per drug/dose/regiment would be useful. For instance, a distribution plot on the costs per drug could be useful.
- It seems that a new drug leads to a big cost increase. Does this include the saved costs of the previous drug? Is this because an alternative treatment is on average more expansive?
- A new dose seems to increase the costs as well. Is this an ‘on average effect’, are there drug dose changes that lead to a decrease of costs?
- It is unclear why the cost of the genetic testing is not part of the reported ~24CAD per patient/year, as the testing is part of the drug therapy cost, and is in question to whether the benefit of testing or not outweighs the costs. Additionally, there should be a statement on the inclusion of related costs such as from counselling and education of Pharmacists and other staff.
- A more thorough discussion and perspective given related literature should be considered. For instance, this study (https://pubmed.ncbi.nlm.nih.gov/29386895/) Assuming a test cost of USD$2,000 for pharmacogenetic testing, the model predicts a savings of USD$3,962 annually per patient with pharmacogenetic-guided medication management.
- It is unclear, whether the calculation includes a projection of costs per year given a continues treatment or whether this is based on the actual prescriptions over the study period.
- It is unclear for how long study participants have been followed
- It is unclear why the study was not designed with ‘placebo’ controls
- Criteria for the inclusion list should be rationalised. Why were patients taking these drugs considered (and not taking other antipsychotics or antidepressants, also such from augmentation strategies)?
The abstract mentions the focus on antidepressant and antipsychotic medications, but the remainder of the article lacks detail and discussion on these drugs and their specific disease domain. I would suggest:
- Give more background on the disease areas and treatment focus in the introduction. There have been various studies and reviews published in this space. Here a shortlist:
- https://www.ncbi.nlm.nih.gov/pmc/articles/PMC7080976/
- https://www.ncbi.nlm.nih.gov/pmc/articles/PMC7518035/
- https://www.karger.com/Article/FullText/492332
- https://pubmed.ncbi.nlm.nih.gov/30135031/
- Table one should be amended with general drug indications and ideally some statistics (e.g. frequencies) on the usage for the 180 study-participants
- In case of drug-switching, to what extend was this discussed and approved by a general practitioner or psychiatrist?
- This following statement does not seem to be true for this study, given a reported 22% ‘Changes to Therapy’, among which it is unclear to what extend can be attributed to antidepressants/antipsychotics. “Two out of three people will need to try multiple/different antidepressants until they find one that works for them”
Given the extensive genetic testing, and annotation of metabolizer phenotypes, it is surprising to see so little analysis on the specific drug-gene-recommendations. Specifically,
- Which drug-gene-phenotype combination has led to most changes to therapy and which did not? From this one could calculate the associated cost per genotype. Are there any surprises or remarks?
- Who received myDNA and who TRS reports? Why where there so many more myDNA tests?
- The time-frame (years of start, enrolment, analytics, etc.) may be given to the reader as this may rationalise the decision for different analytical approaches
- The authors mention a Whole Exome Sequencing approach in Phase 1, which is not clear to how this relates to the presented results. It would be useful to elaborate on the explanations of Phase 1.
- Could the authors elaborate on the performance comparison of the two different genetic tests. Given the differences between the tests, where there any ambiguous treatment decisions that had to be made, based on different genotype calls from where the test phenotypes differed?
- Is there a specific test that is likely more cost-effective, given a cost or performance difference?
- It is unclear if and what fraction of patients have already been on medication prior study enrollment (i.e. prior to what ius described as „To be enrolled in the study a potential participant […] needed to be taking at least 1 of the mental health drugs listed in Table 1.. This is important, as likely those patients with prior treatment are already on a titrated dose and efficacious treatment. In this respect, the authors could provide quantifications on, whether a ‘new’ treatment start, was more likely to have a change of medication outcome, given the PGx results.
- Could the authors provide detail, for which indications the listed drugs have been prescribed, i.e. there are common non-Schizophrenia prescriptions for antipsychotics (e.g. Olanzapine is a used as a sleep aid).
- Were there any adverse reactions observed among those with minor/major considerations and were there any follow-ups on those patients with switched or discontinued treatments? There should be a statement on why such outcomes have not been recorded or discussed in the study.
Although there seems to be strong support for PGx testing by pharmacists and pharmacies, the authors do not discuss the advantages and alternatives to a PGx consultation such as for instance by a general practitioner. In particular, following comments may be reflected:
- In Figure 5, the patient’s perception of the pharmaceutical care is surveyed. However, also a contact address for patients on the consent form is mentioned in the patient’s information video. How many contacts were actually made to this contact address and not referred to the pharmacist? This might be indicative of some hidden concerns by the probands.
- The first conclusion (lines 483 f.) may have limited transferability to other countries and health care systems in which perceptions of patients may differ due to different frameworks – given the publication in an international journal, it should be stated that this conclusion refers to British Columbia.
- Furthermore, the potential of an interprofessional approach should be elaborated in more detail. In particular, the unproductive collaboration between the pharmacist and physician in few cases raises some concerns about the transferability of this approach (lines 351-353). Also, it is mentioned in the introduction (lines 54-56) that this may be a common limitation in a pharmacist-guided PGx based therapy.
Minor comments:
- Typos: supllementary (page 3), Dose increse/decrese (Figure 4), The study tem (Figure 6). It is advised to conduct a general spell checking
- Figure 1 and Table 2 are trivial, have no further implication for the results or interpretation (for instance by discussing the differences between Pharmacies or locations), and would hence suffice in Supplementary
- Figure 2 panels should be aligned with each other though it seems as an effect of typesetting.
- Figure 6: Some annotations seem abrogated (rows: 1, 3, 4?, 6).
- Line 330 states “35A has the same normal metabolizer phenotype“, but is not followed by a reference or own data.
- Table 3 and the associated text should use either of the term “TRS” or “Kailos”, whereas the first term might be preferred.
- Table 3, row 6: Is there a reason why the TRS-genotype is denoted ‘*4/*35A‘ and myDNA ‘*2/*4’, when *35A refers to a subset of the *2 allele (so should the myDNA genotype be rather stated as ‘*4/*2’?)
- Could the authors elaborate and discuss Pharmacogenomic testing in the mental/psychiatric space compared to other disease areas
- Could the authors quantify the following statement: “an overly long time between sample collection and report returns was experienced for the samples collected earliest in the project…”. Additionally, a study timeframe, i.e. the beginning and end of study, would be useful.
Finally, PGx-testing should be discussed more carefully, given examples, which caused more harm than benefit (ie. https://pubmed.ncbi.nlm.nih.gov/28965468/)
Round 2
Reviewer 1 Report
With the respective changes the readability and flow is much better.
Author Response
There are additional minor revisions that we are requesting before acceptance as detailed below. Please address the following additional minor issues:
Please correct the inclusion of 'REF' on line 59 (page 2).—
Added citation PMID: 24763287
Please add citations to the statement on line 67 (page 2)—
Added citation PMID: 31394823
Line 193 (page 5), what is 'METHODS' referring to? please clarify or remove
Changed to (see Expanded Online Methods)
Line279 (page 8), please correct the grammar here.
Changed to- In this study we built on our and others work to further test feasibility of community pharmacogenomic testing.
Please add citations to the statement on line 353 (page 9)-
Added citation
Lines 495-496 (page 15):
More detail is needed to explain and define “annual ongoing treatment costs” versus the “actual prescription over the study period”
added the following sentence
That is to say, the per patient cost of $24.15 represents the average total annual cost increase for each patient’s therapy after implementing the changes. It is not restricted to only the cost of each patient’s therapy for the study period.
Authors more generally need to clarify whether this Drug cost analysis is restricted to mental health medications or any medication; if other medications were included, they should be detailed in the methods section-
Added the following sentence
The restriction to mental health drugs was only for the eligibility to participate. Once a participant was enrolled, we reviewed all their drugs and many of the drug therapy changes that were made were for drugs other than mental health drugs. All drug changes, regardless of therapeutic category, were included in the simple drug costs analysis
Limitations (lines 629-636) should include:
- Focus on mental health medications (line 349)
- Batched samples and up to 6 months between sample collection and return of results (line 459)
Added a sentence - Gene-drug interactions were limited to mental health drugs, and our decision to batch samples for processing slowed the return of results.
